# Resurgent threat in a changing climate: A 20-year bibliometric analysis of global Chikungunya research evolution and pandemic preparedness

Xiaoyi Xu[1], Liming Ye[2], Yawei Liu[1], Xuetao Peng[3]*, Minyi He[4]*

**1** Medical Research Center, The Eighth Affiliated Hospital, Southern Medical University (The First People's Hospital of Shunde), Foshan, China, **2** Sun Yat-sen Memorial Hospital, Sun Yat-sen University, Guangzhou, China, **3** Vice President's office, The Eighth Affiliated Hospital, Southern Medical University (The First People's Hospital of Shunde), Foshan, China, **4** Department of Science and Education, The Eighth Affiliated Hospital, Southern Medical University (The First People's Hospital of Shunde), Foshan, China

* hmy705@smu.edu.cn (MH), 281786009@qq.com (XP)

## Abstract

### Background

Driven by climate change and urbanization, chikungunya virus (CHIKV) has evolved from being a localized pathogen to being a global pandemic threat. This bibliometric analysis maps 20 years (2004–2025) of global CHIKV research to identify priorities and gaps in pandemic preparedness.

### Methods

We analyzed 8,125 publications from the Web of Science using bibliometrix (R) and VOSviewer, examining output trends, collaboration networks, thematic evolution (keyword co-occurrence), and alignment with containment needs.

### Results

Annual publications surged 7-fold after 2013, peaking during epidemics (e.g., Caribbean 2013–2014, Réunion 2024). Research was dominated by the USA, Brazil, France, and India, which formed distinct regional hubs. Thematic clusters revealed rising priorities: vaccine development (proportional growth: +7.1% during 2020–2025; candidates such as VLA1553 emerged) and climate-driven transmission ("environmental health" became central after 2022). Persistent gaps included diagnostic overlap with dengue/Zika and reactive outbreak responses (emergency-focused keywords, e.g., "emergency response and crisis management", declined from 15.5% to 7.2%). International collaborations favored high-income/endemic nations, with limited equity.

**Data availability statement:** The authors confirm that all data underlying the findings are fully available without restriction. All relevant data are within the paper and its Supporting Information files.

**Funding:** This work was supported by the Guangdong Provincial Education Department Scientific Research Platforms and Projects (Grant number 202430-8 to M.H.); the Guangdong Provincial Medical Research Fund (Grant number B2025301 to M.H.); the Foshan Science and Technology Bureau (Grant number 2420001003729 to M.H.); and the Southern Medical University (Grant number ZL2023113 and ZL2024135 to M.H.). The funders had no role in study design, data collection and analysis, decision to publish, or preparation of the manuscript. No authors received a salary from any of the funders.

**Competing interests:** The authors have declared that no competing interests exist.

## Conclusion

While CHIKV research increasingly addresses vaccines and climate drivers, critical weaknesses remain in terms of equitable collaboration, climate-adaptive surveillance, and integrated interventions. We advocate for predictive modeling, rapid-response vaccine platforms, and embedding CHIKV preparedness within climate-resilient health policies to transform reactive efforts into sustained pandemic resilience.

### Author summary

Chikungunya, a mosquito-borne virus that causes severe joint pain and fever, has exploded from a regional concern into a global threat, fueled by climate change and urbanization. To understand how the world is preparing, we analyzed 20 years of global scientific research (2004–2025). Our findings reveal a research field intensely responsive to outbreaks, with publications surging 7-fold during major epidemics. The scientific focus has matured from simply describing outbreaks to proactively developing vaccines and understanding how climate change expands the range of mosquitoes. However, global research efforts are uneven. While the US, France, Brazil, and India have formed strong collaborative hubs, many affected regions, particularly in Africa, are left out. There are critical gaps in rapid outbreak response and equitable access to new tools, such as vaccines. This analysis provides a roadmap: To increase pandemic resilience, we must promote more equitable global partnerships, integrate climate predictions into health planning, and ensure that scientific advances are translated into real-world protection for vulnerable communities.

## 1. Introduction

Chikungunya virus (CHIKV), an arthropod-borne alphavirus transmitted primarily by *Aedes aegypti* and *Aedes albopictus* mosquitoes, has emerged from being a historically localized pathogen to become a pervasive global health challenge [1]. Characterized by acute febrile illness and debilitating polyarthralgia, which can persist for years as chronic arthritis, chikungunya imposes substantial morbidity and socioeconomic burdens in affected regions [2]. Since its re-emergence in 2004–2005 (Indian Ocean lineage), CHIKV has demonstrated explosive epidemic potential and has spread to more than 100 countries across tropical and subtropical zones [3]. In Brazil, the hardest-hit region, 41% of the country's cases were reported in 2018 alone [4]. Critically, climate change and unplanned urbanization have accelerated the expansion of competent mosquito vectors into new geographic areas, facilitating autochthonous transmission in temperate regions of Europe and North America [5]. This aligns with the epidemiological trajectory of CHIKV, which, over the past decade, has experienced repeated resurgences in India [6], Brazil [7], and parts of Africa [8]—regions characterized by endemic transmission cycles, increasingly permissive

climatic conditions, and expanding vector niches. Particularly concerning is the virus's ability to adapt to *Aedes albopictus*, a more temperate-tolerant mosquito than *Aedes aegypti*, which facilitates transmission in subtropical and even temperate zones [9]. Autochthonous CHIKV transmission has been confirmed in southern France, Italy, and even Spain, highlighting Europe's vulnerability [10]. A similar trend is observed in the United States, where the geographic range of Aedes mosquitoes has expanded to more than 30 states [11]. These changes, driven by global warming, urbanization, and international mobility, have created an ecological setting favorable for the reestablishment of the virus. This ecological shift, coupled with rising global mobility, has transformed chikungunya from a sporadic tropical disease to a persistent pandemic threat. Since August 2024, Réunion Island (France) has experienced widespread chikungunya transmission, and the number of locally acquired cases has increased in Mayotte [5]. Although chikungunya occurs annually in many parts of the world, Indian Ocean islands have not faced major outbreaks for nearly two decades. As of 4 May 2025, Réunion has reported more than 47,500 cases and 12 related deaths, while 116 cases were recorded in Mayotte [12]. In response, both territories have implemented enhanced surveillance, vector-control campaigns, and targeted vaccination. Nonetheless, further chikungunya activity is anticipated across Indian Ocean islands.

For decades, understanding CHIKV research priorities has relied on narrative reviews and isolated outbreak reports [6]. However, the exponential growth of publications, coupled with complex interactions between virological, epidemiological, and environmental factors, has made it increasingly challenging to holistically map knowledge trajectories and emerging themes. Bibliometric analysis offers a powerful alternative, using quantitative characteristics (e.g., keywords, citations, collaborations) to objectively identify research contributions, evolutionary trends, and future hotspots within a field [13].

Network analysis of keywords is particularly valuable for delineating conceptual relationships. When terms consistently cooccur in publications, for example, "climate change" and "vector competence", they reveal intrinsic thematic linkages and emerging interdisciplinary foci. Furthermore, sustained increases in keyword frequency over time can signal the development of research frontiers with high translational potential. No study has systematically mapped the 20-year evolution of CHIKV research, a critical gap given its escalating pandemic threat.

Thus, this scientometric study has the following aims: (1) To quantify global research output and collaboration patterns for 2004 to 2025; (2) To identify thematic clusters and their temporal evolution using co-word bi-clustering; (3) To assess the alignment between research foci and pandemic containment needs in an era of climate-driven transmission.

This bibliometric analysis synthesizes 20 years of global research evolution (2004–2025) to map scientific priorities, collaboration patterns, and emergent themes, providing a strategic framework to align research investments with pandemic containment needs in an era of climate health crises.

## 2. Methods

### 2.1. Data source and retrieval strategy

The Web of Science Core Collection (WoSCC) was exclusively queried on July 26, 2025, which yielded 9,440 initial records. The search strategy targeted publications from 2004 to 2025 using the following Boolean query: TS = ("chikungunya virus" OR "chikungunya fever" OR chikungunya OR "chik virus" OR CHIKV) AND PY = (2004-2025).

### 2.2. Screening and inclusion criteria

Three sequential filters were applied: a document type restriction to articles or reviews (retaining 8,131 records) and duplicate removal using DOI-based deduplication in EndNote X20, supplemented by manual verification of author-year-title mismatches. We identified and removed duplicates using a two-step process: first, automated deduplication in EndNote X20 based on DOI and title matching; second, manual verification to resolve any discrepancies (e.g., variations in author names or publication years). This process identified 6 duplicate records, which were excluded, resulting in a final analytic dataset of 8,125 unique publications (Fig 1). Full metadata, including titles, abstracts, keywords, affiliations, citations, journals, and funding, were exported in plain-text format (S1 Appendix).

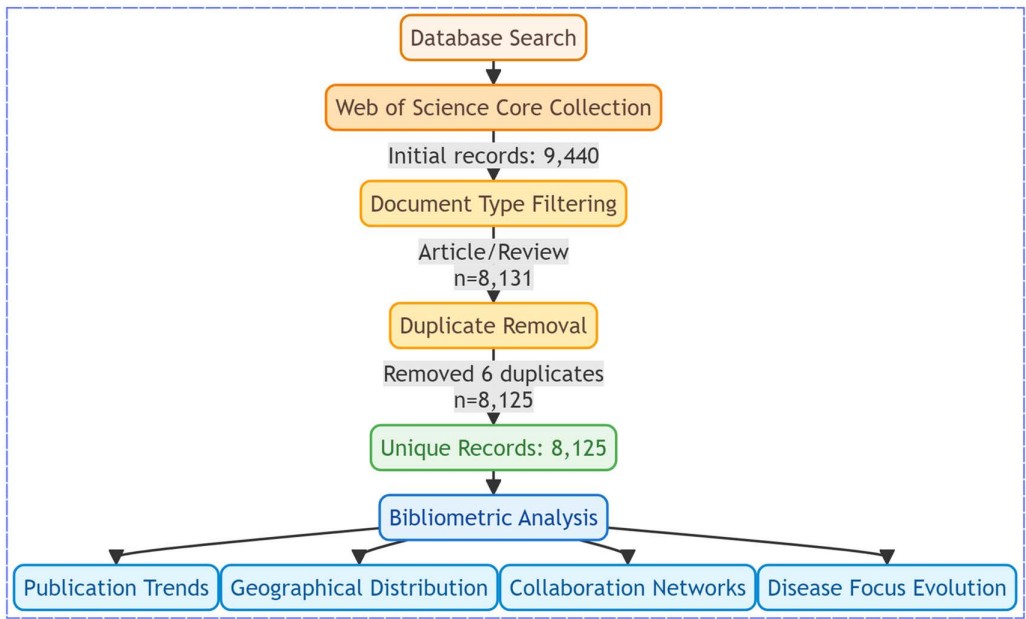

**Fig 1. 1 Flowchart of the screening process.**

### 2.3. Analytical framework

A bibliometric analysis was conducted employing R 4.4.3 (utilizing the bibliometrix package) and VOSviewer 1.6.20, examining six dimensions: Publication Trends, which included annual and cumulative output (from 2004 to 2025), visualized through line graphs; Geographical Distribution, featuring country-level contributions with fractional counting and collaboration networks for countries with at least 160 publications; Journal Metrics, encompassing the Journal Impact Factor (JIF 2024), CiteScore, and quartile rankings, with scatter plots comparing JIF against publication volume; Collaboration Networks, focusing on institutional coauthorship (with a minimum of 90 documents per organization); Thematic Evolution, involving keyword co-occurrence analysis (with a minimum occurrence threshold of 50) and timeline clustering; and Disease/Outbreak Focus, entailing content analysis of abstracts for clinical themes, such as "chronic arthralgia," "neonatal transmission," and "vector competence," and public health interventions, such as "vaccine development" and "serosurveillance".

### 2.4. Data standardization

To ensure consistency, keyword normalization was performed, whereby variants, such as "CHIKV" and "chikungunya virus", were merged, and generic terms, such as "review", "article" and "study", were excluded. Affiliation harmonization involved standardizing institutional names; for instance, "CDC" was changed to "Centers for Disease Control and Prevention," and "Inst Pasteur" was changed to "Institut Pasteur" (S1 Table).

### 2.5. Data analysis and visualization

Bibliometric data were analyzed using two complementary tools, VOSviewer (version 1.6.20) and R (version 4.4.3, with the 'circlize', 'ggmap', 'RColorBrewer', 'tidyverse', 'readxl', 'ggalluvial', 'viridis', 'dplyr', 'ggplot2' and 'writexl' packages).

Publication trend analysis: Annual publication counts and citation metrics were calculated using 'circlize', 'ggmap', 'RColorBrewer', 'tidyverse', and 'readxl' to characterize temporal trends in CHIKV research. The base map layer depicting

country borders was generated using the map_data('world') function from the ggplot2 package in R (version 4.4.3), which utilizes publicly available geographic data from the maps package (source: https://cran.r-project.org/web/packages/maps/). These data are derived from public domain sources, including the U.S. Census Bureau, and are compatible with the CC BY 4.0 license.

Collaboration network analysis: Institutional and national collaboration networks were visualized in VOSviewer using the "coauthorship" module, with node size representing publication volume and edge thickness representing collaboration frequency.

Thematic evolution analysis: Keyword co-occurrence networks (top 50 keywords by frequency) were constructed to identify thematic clusters (e.g., "Epidemiology", "Vaccine Development") and their temporal changes ("2016-2017" vs. "2018-2019" vs. "2020-2021" vs. "2022-2023" vs. "2024-2025") using 'tidyverse', 'readxl', 'ggalluvial', 'viridis', 'dplyr', 'ggplot2' and 'writexl'.

Visualization quality control: All figures were optimized for clarity (e.g., adjusting node spacing in networks and standardizing color scales for thematic clusters).

## 3. Results

### 3.1. Analysis of publication volume

A total of 8,125 publications on chikungunya research were analyzed, spanning the years from 2004–2025. The annual publication output exhibited three distinct phases (Fig 2 and S2 Table): (1) Foundational Phase (2004–2013) Although the chikungunya virus was first identified in 1952, research activity remained limited until the large-scale reemergence of CHIKV on Indian Ocean islands (e.g., Réunion) from 2004–2005. This outbreak, caused by a new lineage (Indian Ocean lineage) with enhanced transmission by *Aedes albopictus*, marked the beginning of the modern era of CHIKV research. During this phase, the annual output averaged 109 publications per year, increasing from a low of 7 publications in 2005 to a peak of 230 in 2013. This period established the foundational research framework for subsequent studies but accounted for only 13.4% (n = 1,091) of the total output. (2) Epidemic-Driven Growth Phase (2014–2020): Annual publications increased 7-fold above pre-2014 levels, from 273 (2014) to 768 (2020). This explosive growth coincided with the 2013–2014 Caribbean epidemic and subsequent global spread. The steepest increase occurred between 2014 and 2017 (+323 publications; +118% growth) and accounted for 49.3% (n = 4,003) of all publications. (3) Post-Pandemic Adjustment Phase (2021–2025): Output stabilized at elevated levels but declined from peak values: 2021: n = 734 (4.6% decline from the peak), 2022: n = 710 (8.2% decline), 2023–2024: stabilized near 2018–2019 levels (~600 publications/year); the 2025 data (n = 394) are provisional (current to July 26, 2025) and expected to increase.

### 3.2. Analysis of national publication counts and collaboration networks

**3.2.1. National contributions.** The 8,125 publications originated from 20 core countries (Fig 2B and S2 Table). The United States (USA) dominated with 2,626 publications (32.3% of the total), followed by Brazil with 1,240 publications (15.3%), France with 1,067 (13.1%), and India with 1,011 (12.4%). The United Kingdom (UK; n = 628), Germany (n = 436), and Australia (n = 400) formed a secondary tier, while countries such as Thailand and Mexico (n = 239) demonstrated significant output despite lower overall volumes. It is important to note that this distribution may be influenced by the fact that most relevant research is published in English: Brazil (with significant Portuguese-language research), India (regional languages), and China (Mandarin) likely have produced additional non-English publications that are not captured here, which may underestimate their actual research output and could affect interpretations of international collaboration patterns.

**3.2.2. Global collaboration patterns.** International collaborations established a tripartite network anchored by three hubs (Fig 2C and S2 Table): the Americas Hub (USA-Brazil), featuring the strongest bilateral link (collaboration

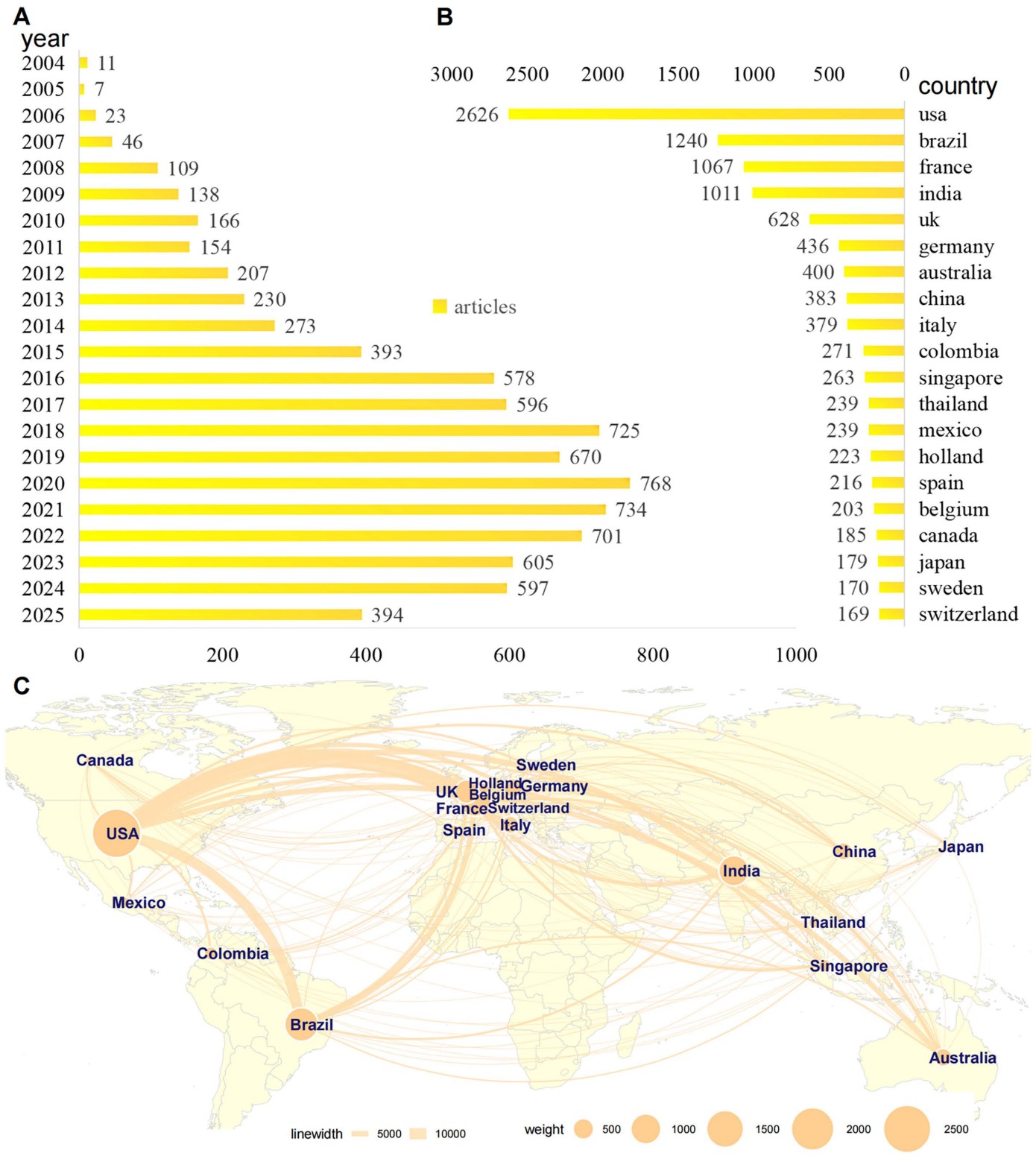

**Fig 2. Global Research Output and International Collaboration Networks (2004–2025). (A)** Trends in the number of articles published from 2004 to 2025. **(B)** Publication output from the top 20 contributing countries/regions. **(C)** A collaborative network map of the top 20 contributing countries/regions. Base map data sourced from the maps package (source: https://cran.r-project.org/web/packages/maps/) in R, which uses public domain geographic data.

strength: 9,455), along with integrated Canada (n = 185), Mexico (n = 239), and Colombia (n = 271); the Europe Hub (France-UK-Germany), characterized by the highest multilateral density (France-UK: 3,674; France-Germany: 2,615), extending to Belgium (n = 203), Switzerland (n = 169), and Sweden (n = 170); and the Asia-Pacific Hub (India-Singapore), which is focused on outbreak regions (India-Thailand: 733; Singapore-Australia: 901), with Japan (n = 179) and China (n = 383) showing limited external linkages.

Notable Dynamics: Trans-Atlantic Bridge: The collaboration between the USA and France (strength: 14,181) surpassed all intra-European partnerships. Endemic-Developed Nation Ties: The partnerships between Brazil and France (5,104) as well as Brazil and the UK (2,852) were stronger than Brazil's regional collaborations were. Isolation Trend: At only 3.4% of the USA's partnership strength, China displayed the lowest collaboration-to-volume ratio.

### 3.3. Analysis of related institutions

The institutional collaboration network among the top 20 organizations (Fig 3 and S3 Table) revealed distinct geographic clusters and transcontinental partnerships. Institutions were grouped into five color-coded clusters reflecting primary national affiliations: Cluster 1 (USA; red), Cluster 2 (Brazil; blue), Cluster 3 (France; green), Cluster 4 (UK; purple), and Cluster 5 (Other Countries; orange).

Dominant Hubs:

Institut Pasteur (France) emerged as the most influential node, producing the highest volume of publications (n=333) and maintaining its strongest collaboration with the University of Texas Medical Branch (USA) (collaboration strength=1,180). Fiocruz Minas Gerais (Brazil) (n=302 publications) anchored Cluster 2, with intense domestic ties to the University of São Paulo (strength=380).

Transatlantic Synergies:

The strongest cross-cluster collaborations linked France (Cluster 3) and the USA (Cluster 1) (Institut Pasteur - University of Texas Medical Branch: 1,180); Brazil (Cluster 2) and the USA (Fiocruz Minas Gerais - University of Texas Medical Branch: 446); and France and the UK (Institut Pasteur - University of Oxford: 321).

Cluster-Specific Dynamics:

USA (Cluster 1): Characterized by high-volume institutions (University of Florida, CDC) with diffuse global partnerships.

Other Countries (Cluster 5): Multinational entities (e.g., Ministry of Health) served as bridges between clusters, particularly those linking Asia-Pacific and European hubs.

Collaboration Asymmetry:

The University of Texas Medical Branch (USA) acted as the central global connector, maintaining the top three international partnerships (France, Brazil, and the UK), while Chinese institutions exhibited minimal external linkages.

The network illustrates a core-periphery structure in which French, Brazilian, and U.S. institutions form the collaborative backbone, driven by endemic regional expertise (Brazil) and high-resource research capacity (USA/France). This pattern mirrors the tripartite national alliances identified in Section 3.2, confirming that macrolevel partnerships manifest institutionally.

### 3.4. Analysis of journal distribution and impact

Dominant Publication Venues: A total of 3,015 articles were published in the top 20 journals specializing in chikungunya research (Fig 4 and S4 Table). PLoS Neglected Tropical Diseases emerged as the primary platform, publishing 562 articles (18.6% of the subset). This was followed by Viruses-Basel (314; 10.4%) and PLoS One (289; 9.6%). Research Field Clustering: Four thematic domains were identified: Parasitology & Tropical Diseases, with core journals including PLoS NTDs (562), Parasites & Vectors (232), and AJTMH (195); Virology & Antiviral Research, led by journals such as Viruses-Basel (314), Journal of Virology (185), and Antiviral Research (84); Infectious Diseases & Public Health, anchored by Emerging Infectious Diseases (91) and BMC Infectious Diseases (70); and Multidisciplinary Sciences, dominated by PLoS

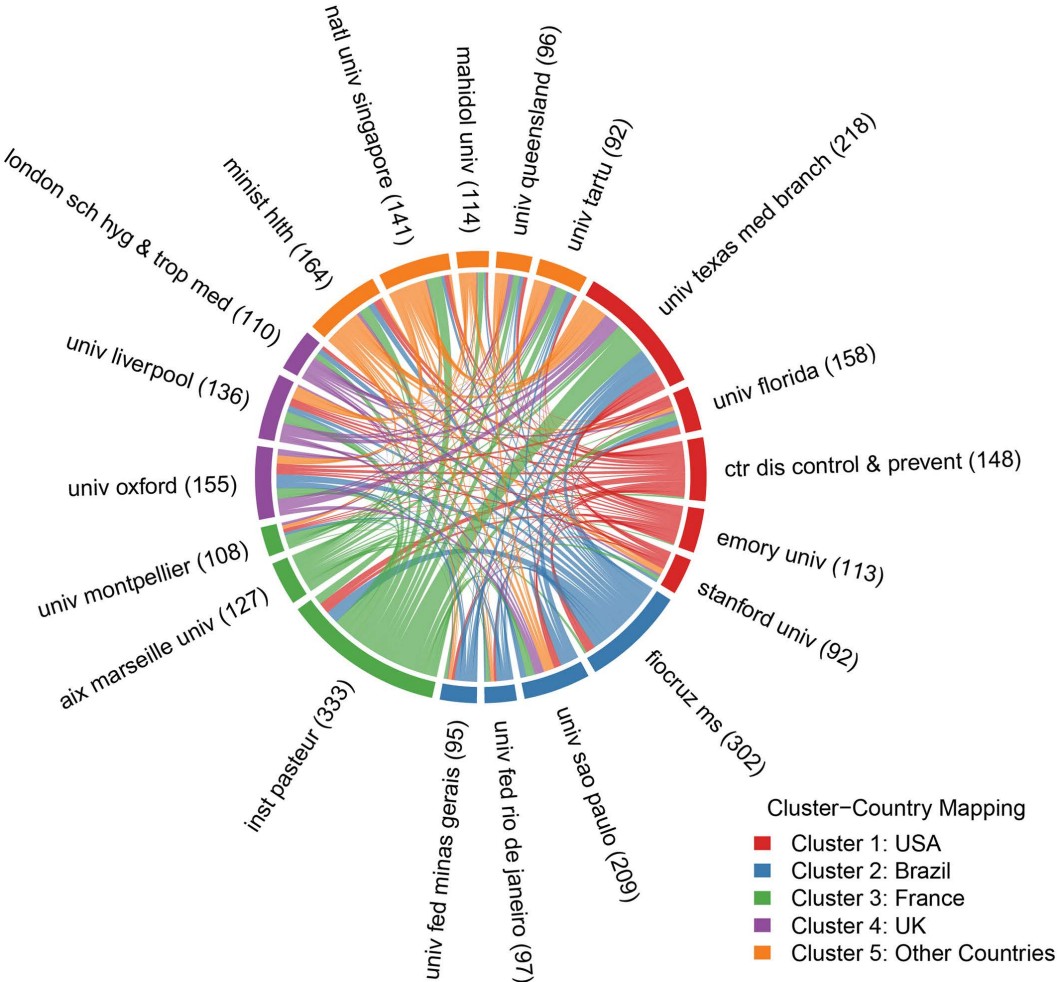

**Fig 3. A collaborative network map of the top 20 contributing organizations.**

One (289) and Scientific Reports (182). A comparison between impact factor and productivity revealed two significant patterns: High-Impact Specialization is exemplified by journals such as Emerging Infectious Diseases, with an impact factor (IF) of 6.6, and PLoS Pathogens, with an IF of 4.9, which publish fewer articles but of high impact. Antiviral Research, with an IF of 4.0, maintains a moderate level of productivity, having published 84 articles. Field-Specific Gradients are evident, with virology journals having the highest mean IF at 3.9 ± 0.6, while tropical medicine journals, with a lower IF of 2.5 ± 0.7, exhibit a higher output.

### 3.5. Analysis of cocited articles

The co-citation network of the 20 most influential articles (Fig 5 and S5 Table) reveals four distinct research clusters: Chikungunya Virus Research (blue, 7 nodes), Zika Virus Research (orange, 5 nodes), Arbovirus Biology and Vector Control (green, 4 nodes), and Dengue and Arbovirus Interactions (red, 4 nodes). These clusters are specialized in chikungunya research and interconnected through key methodological relationships. Nodes represent highly cited articles (with size scaled by citation count), while edges denote co-citation strength (with width proportional to weight).

Key Publications:

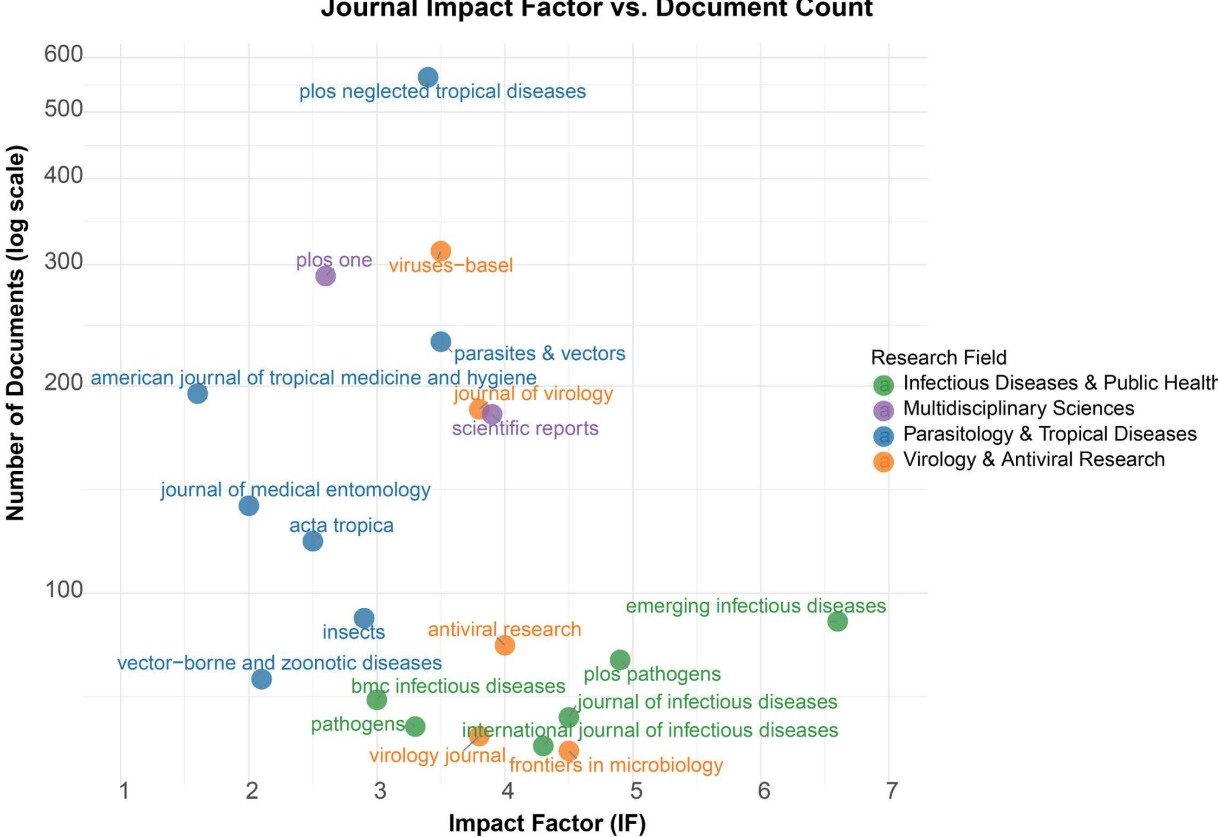

**Fig 4. Scatter plot distribution of IF and document count for the top 20 journals.**

The Chikungunya cluster was anchored by Tsetsarkin et al. (2007, PLoS Pathogens) (805 citations), who reported that adaptive mutations enhance viral fitness in Aedes mosquitoes. The Zika cluster was characterized by Bhatt et al. (2013, Nature) (728 citations), who mapped global epidemic risk through geospatial modeling. A highlight of the Arbovirus Biology cluster was Strauss & Strauss (1994, Microbiology Reviews) (469 citations), a foundational review on alphavirus structure. The Dengue cluster was centered on a study by Rezza et al. (2007, The Lancet) (780 citations), which links the emergence of chikungunya to the emergence of *Aedes albopictus* in Europe.

Critical Co-citation Relationships:

The three strongest co-citation links (edge weight >200) revealed several interdisciplinary bridges. Schuffenecker et al. (2006) - Tsetsarkin et al. (2007) (weight=315) connects chikungunya phylogenetics (African/Asian strains) with evolutionary mechanisms of mosquito adaptation. Tsetsarkin et al. (2007) - Vazeille et al. (2007) (weight=248) unites viral evolution studies with vector competence experiments in *Aedes aegypti*. Rezza et al. (2007) - Tsetsarkin et al. (2007) (weight=241) bridges dengue/chikungunya cocirculation epidemiology with molecular determinants of transmission.

Topological Insights:

Cluster 1 (Chikungunya) exhibited the highest internal cohesion, with high numbers of citations (≥390) occurring within the cluster. Cross-cluster synergy was prominent between Clusters 1 (Chikungunya) and 4 (Dengue), reflecting shared virological mechanisms and vector ecology. Citation asymmetry: Early foundational works (e.g., Robinson, 1955) had fewer citations but functioned as bridges between clusters.

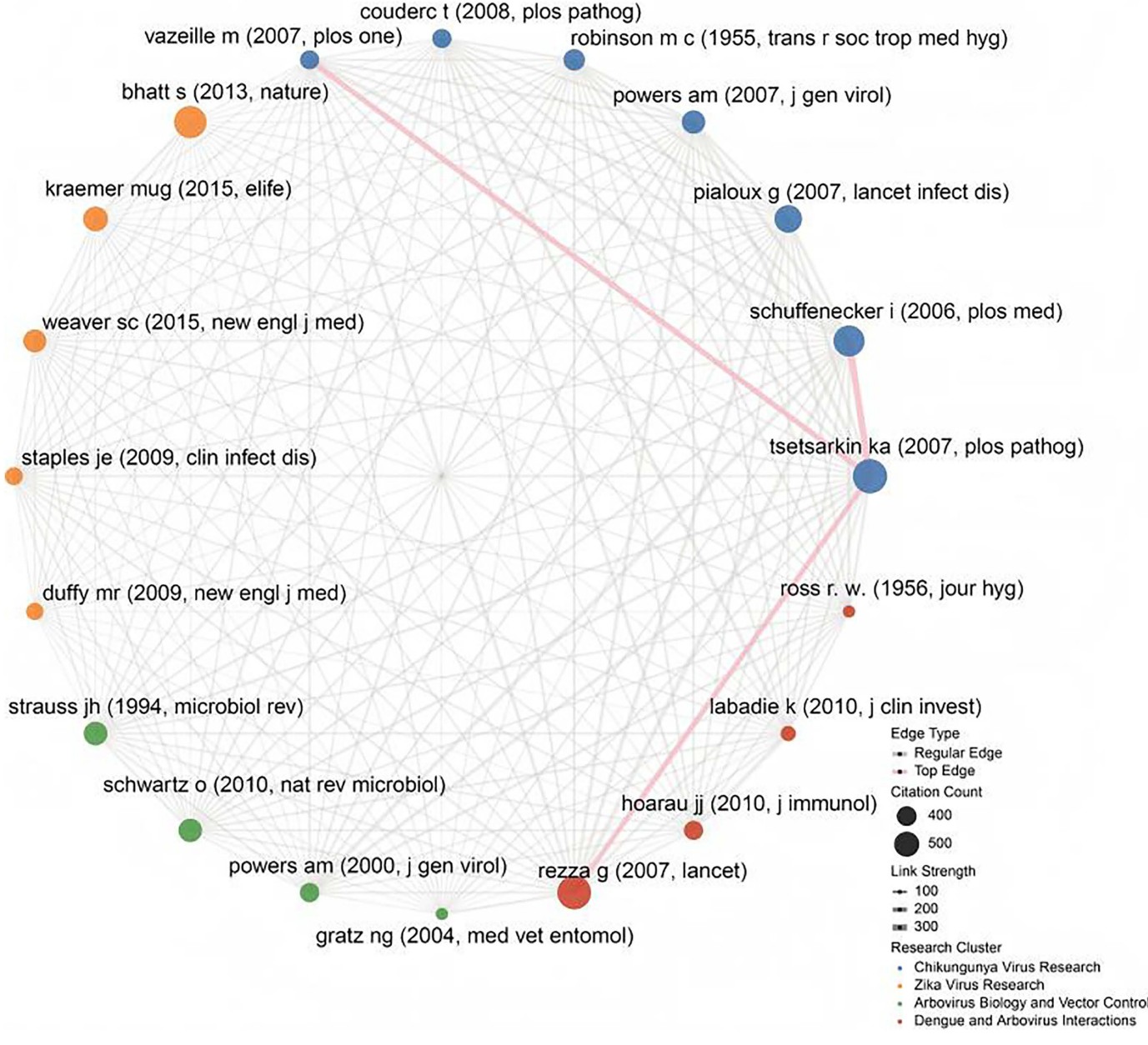

**Fig 5. The co-citation network of the 20 most frequently cited articles.**

The network emphasizes Chikungunya research as the dominant paradigm, with Zika studies emerging as a distinct but interconnected field. Strong connections between dengue and chikungunya research highlight synergies in arbovirus pathogenesis and control.

### 3.6. Analysis of Chikungunya research-related social networks

Keyword co-occurrence analysis of the top 51 terms (2004–2025, occurrence≥50) revealed five distinct thematic clusters (Fig 6A and S6 Table), reflecting the multifaceted nature of chikungunya research: (1) Core Pathogens & Diseases (red),

(2) Transmission Vectors (blue), (3) Transmission Dynamics & Epidemic Characteristics (green), (4) Control Strategies & Interventions (purple), and (5) Influencing Factors & Host Response (orange). Below, we focus on the evolutionary trends of the two key clusters with the most pronounced temporal shifts in proportional focus.

**3.6.1. Core pathogens & diseases (red cluster).** This cluster (n=5,002 total occurrences) focused on pathogen taxonomy and disease entities and featured 20 virus-specific keywords (excluding symptom terms, such as "arthralgia"). Temporal analysis (2016–2025; Fig 6B and S6 Table) revealed a significant upward trend in the proportional emphasis on "arboviruses" and "chikungunya". From 2020–2021 to 2024–2025, the proportion of "arboviruses" within the cluster increased by 3.8 percentage points (from 22.2% to 26%), reinforcing its position as the dominant umbrella term for vector-borne viral research. Moreover, "chikungunya" (including "chikungunya virus") experienced a proportional growth of 5.2 percentage points (from 27.6% to 32.8%), reflecting its increased priority as a distinct pathogen within broader arboviral studies. Notably, cocirculating arboviruses exhibited divergent proportional trends: "dengue" remained stable (14.4% in 2020–2021 versus 13.6% in 2024–2025), whereas "Zika virus" decreased by 4.5 percentage points (from 10.4% to 5.9%), aligning with reduced outbreak-driven research attention. This shift highlights the emergence of CHIKV as a primary focus within arboviral research.

**3.6.2. Control strategies & interventions (purple cluster).** This cluster (n=817 total occurrences) encompassed prevention and mitigation approaches, with 9 key terms. Evolutionary analysis (2016–2025; Fig 6C and S6 Table) highlighted significant proportional growth in "vaccine" and "public health" interventions. The percentage of "vaccines" within the cluster increased by 8.1 percentage points from 2020–2021 (14.2%) to 2024–2025 (22.3%), driven by advancing clinical trials and candidate development. Moreover, "public health" showed a proportional increase of 3.9 percentage points (from 7.4% to 11.3%), reflecting an expanded focus on implementation frameworks and equitable access. In contrast, "antiviral" research declined (32.3% to 22.3%), whereas "insecticide resistance" remained stable (6.5% vs. 7.0%), suggesting a shift toward proactive prevention rather than chemical vector control. Moreover, "mosquito/vector control" maintained its foundational role and even slightly increased (14.8% to 16.4%).

**3.6.3. Vaccine-related keyword evolution (2016–2025).** To further dissect the proportional growth in vaccine research, we analyzed 74 vaccine-specific keywords (2016–2025), categorized into five subthemes (Fig 7A and S7 Table). "Vaccine composition and types" emerged as the fastest-growing subtheme, with its proportion within vaccine-related research increasing by 4.6 percentage points from 2016–2017 (11.2%) to 2024–2025 (15.8%), surpassing other subthemes in terms of relative growth. Network analysis of 12 representative composition/type keywords (Fig 7B and S7 Table) revealed temporal diversification: early terms (2016–2018) focused on traditional platforms (e.g., "subunit vaccines" and "live attenuated vaccines"). By 2025, novel candidates "VLA1553" and "IXCHIQ" (a live-attenuated vaccine and its brand name) first appeared, coinciding with late-stage clinical trial progress and highlighting the linkage between research focus and development milestones.

**3.6.4. Public health-related keyword evolution (2016–2025).** To further explore the expanding focus on public health interventions in chikungunya research, we analyzed 61 public health-specific keywords (2016–2025), categorized into four subthemes (Fig 8A and S8 Table): Disease Surveillance and Risk Assessment, Disease Prevention and Control, Emergency Response and Crisis Management, and Health Systems and Global Eco-Health. Disease Prevention and Control emerged as the most dynamically growing subtheme, with its proportion increasing by 5.7 percentage points from 2018–2019 (32.5%) to 2024–2025 (38.2%), reflecting intensified efforts to translate research into actionable prevention strategies. Concurrently, Health Systems and Global Eco-Health increased by 7.9 percentage points (30.3% to 38.2%), underscoring the growing recognition of chikungunya as a transboundary threat requiring integrated, planetary health frameworks.

In contrast, Emergency Response and Crisis Management declined notably (15.5% in 2016–2017 to 7.2% in 2024–2025), suggesting a transition from reactive outbreak containment to preemptive preparedness. Disease Surveillance and Risk Assessment remained relatively stable (19.9% to 16.5%), maintaining its role as a foundational pillar but with increased emphasis on data-driven risk modeling.

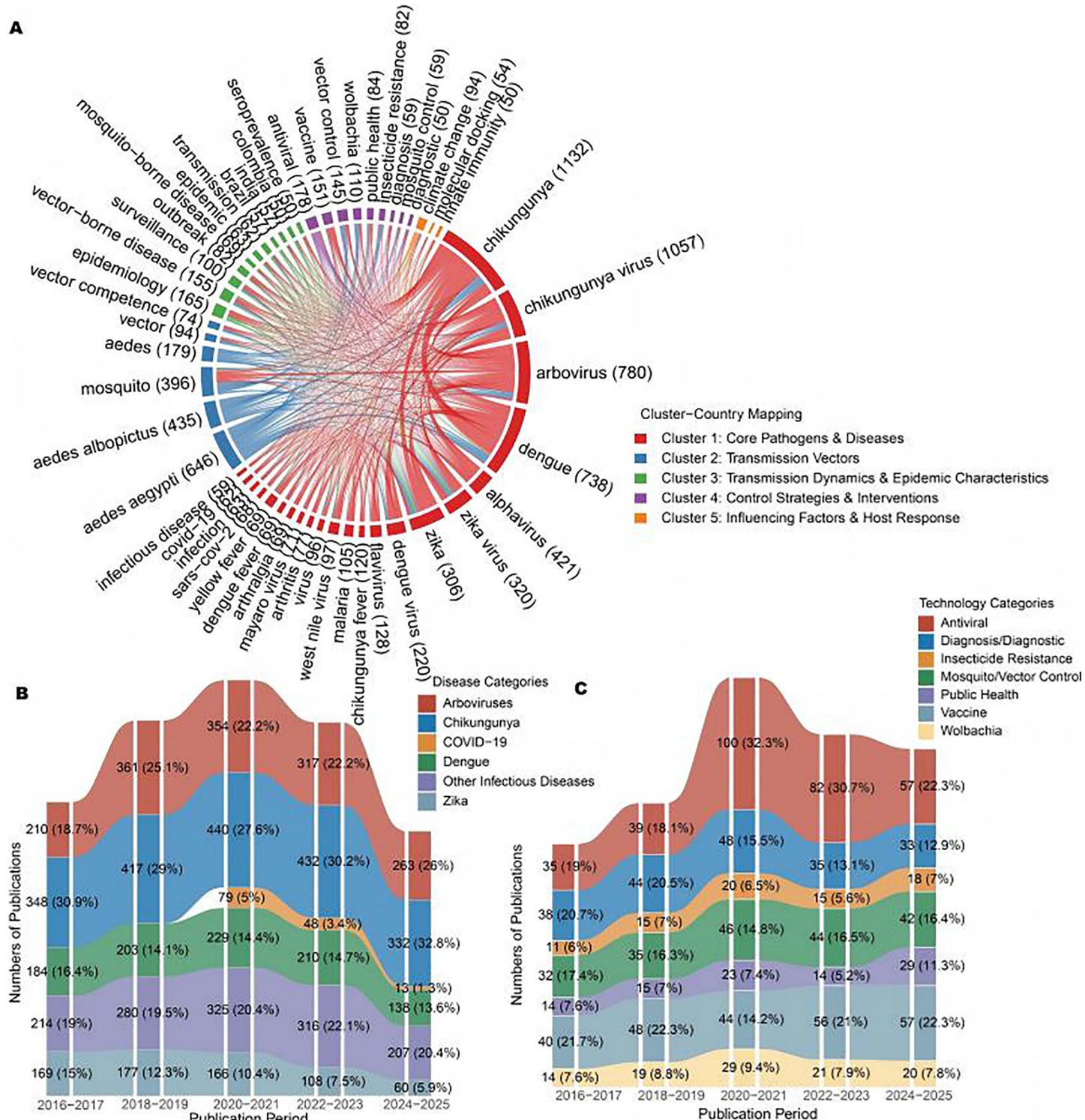

**Fig 6. Keyword Co-occurrence Network and Thematic Evolution in Chikungunya Research (2004–2025).** A. The network of top 50 keyword co-occurrence in chikungunya research (2004-2025). B. Evolution of core pathogens & diseases (2016-2025). C. Evolution of control strategies & interventions (2016-2025).

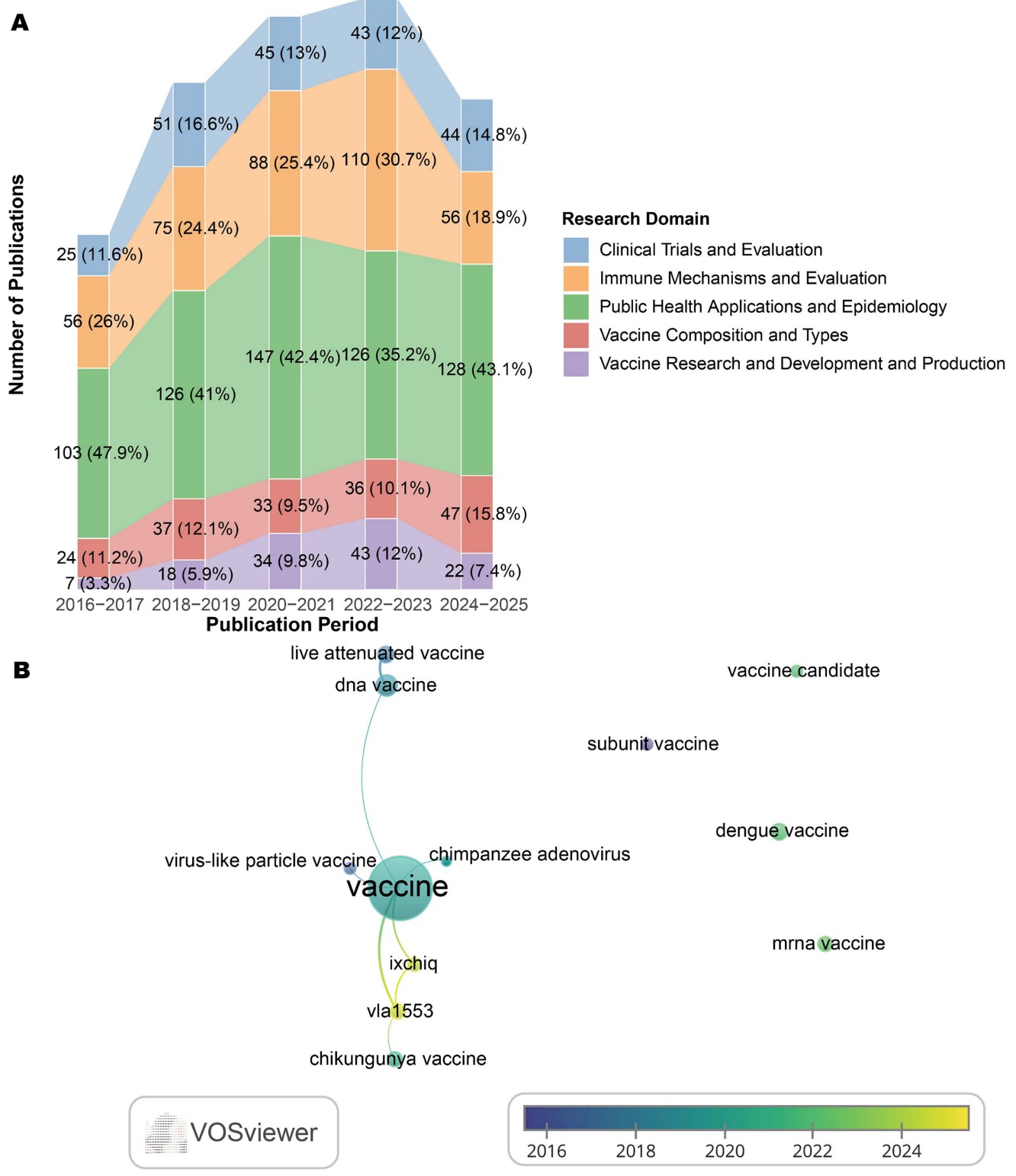

**Fig 7. Evolution of Vaccine-Related Research Themes in Chikungunya (2016–2025).** A. The evolution of all vaccine-related keywords in chikungunya research (2016-2025). B. Evolution of keywords in vaccine composition and types (2016-2025).

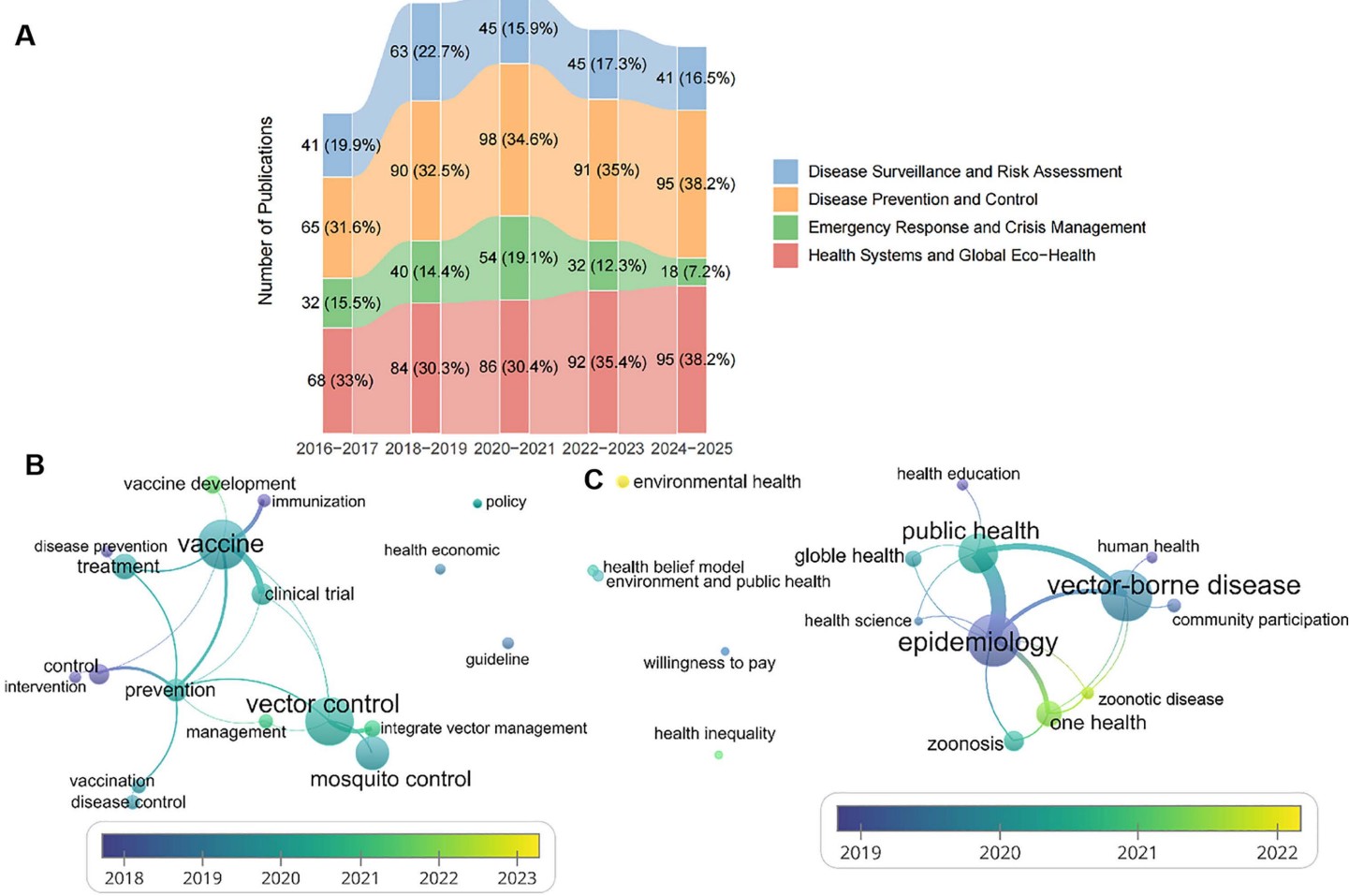

**Fig 8. Evolution of Public Health-Related Research Themes in Chikungunya (2016–2025).** A. The evolution of all public health-related keywords in chikungunya research (2016-2025). B. Evolution of keywords in disease prevention and control (2016-2025). C. Evolution of keywords in health systems and global eco-health (2016-2025).

Disease Prevention and Control Subtheme (Fig 8B and S8 Table)

Network analysis of 18 representative keywords in the Disease Prevention and Control subtheme revealed a temporal shift toward integrated, multilayered strategies. Early keywords from 2016 to 2020 focused on traditional vector control measures, such as "mosquito control", "vector control", and "intervention". A significant shift occurred after 2022, with the emergence of "vaccine development" as a prominent keyword. This shift aligns with the advancement of late-stage vaccine candidates (e.g., VLA1553 and IXIARO) and reflects the integration of immunization into core prevention frameworks.

Health Systems and Global Eco-Health Subthemes (Fig 8C and S8 Table)

Network analysis of 16 keywords in Health Systems and Global Eco-Health revealed a growing focus on structural resilience and environmental determinants. Prior to 2022, the subtheme was dominated by terms related to global collaboration, such as "global health" and "international intervention", as well as those pertaining to health equity, including "health inequality" and "equitable access".

A striking development after 2022 was the emergence of "environmental health" as a central keyword, which coincided with intensified research on climate-driven vector expansion.

## 4. Discussion

### 4.1. Interpreting two decades of CHIKV research through a bibliometric lens

This bibliometric analysis delineates the evolution of global CHIKV research from 2004 to 2025, revealing a field intensely responsive to outbreak dynamics yet one facing critical challenges in equity, translation, and climate adaptation. Our findings quantify a 7-fold surge in annual publications after 2013 (Section 3.1, Fig 2A and S2 Table), directly mirroring the rapid global spread of the virus and establishing chikungunya as a persistent pandemic threat. This growth was not merely quantitative; keyword evolution revealed a proportional increase of 5.2 percentage points for "chikungunya" within the core pathogen cluster (Section 3.6.1, Fig 6B and S6 Table), signaling its maturation from a niche subject to a central focus within arbovirology. The convergence of increasing output and thematic focus underscores the scientific community's increasing response to the expanding burden of CHIKV.

### 4.2. Thematic evolution: From reactive outbreak science to proactive preparedness

This maturation is evidenced by the significant proportional growth (8.1 percentage points, from 14.2% to 22.3%) of the keyword "vaccine" within the Control Strategies & Interventions cluster from 2020–2021 to 2024–2025 (Section 3.6.2, Fig 6C and S6 Table). The temporal diversification in vaccine research is highlighted by the emergence of specific candidate names, such as "VLA1553" and "IXCHIQ", in keyword networks by 2025, aligning with late-stage clinical progress (Section 3.6.3, Fig 7B and S7 Table). This trend aligns with tangible progress in the field, exemplified by the first regulatory approvals of a chikungunya vaccine (VLA1553, marketed as IXCHIQ) by the U.S. FDA in late 2023 and other agencies thereafter, with initial deployment strategies currently targeting high-risk groups in endemic regions [14]. Despite this milestone having been achieved, the path to widespread vaccine adoption may face significant barriers. Lessons from COVID-19 and other vaccine introductions highlight potential hurdles, including vaccine hesitancy, logistical challenges related to cold-chain requirements for live-attenuated vaccines, cost-effectiveness considerations in resource-limited settings, and the need for robust pharmacovigilance systems [15]. Proactively addressing these implementation challenges is essential for ensuring equitable access and achieving successful integration of chikungunya vaccines into public health programs in endemic regions.

Conversely, the proportional decline in Emergency Response and Crisis Management keywords (-8.3%) (Section 3.6.4, Fig 8A and S8 Table) is a critical finding. Although this shift may reflect welcome progress toward proactive preparedness, it could also reflect a troubling decline in the priority given to reactive outbreak capacity—a capability that remains indispensable in light of the virus's unpredictable nature. Bibliometric data alone cannot distinguish between these interpretations; this decline may reflect a maturation of the field beyond initial outbreak response reports, a shift in terminology, or a genuine gap. Therefore, we interpret this trend not as a definitive measure of capacity loss but as a signal warranting further qualitative investigation into whether research is adequately supporting frontline outbreak response needs.

### 4.3. Collaboration networks: Strength does not equate to equity

Our network analyses reveal a robust but imbalanced global research architecture. The tripartite network anchored by the USA, France, Brazil, and India (Section 3.2.2, Fig 2C and S2 Table) and reinforced at the institutional level (Section 3.3, Fig 3 and S3 Table) has driven high-output science. However, the low collaboration-to-volume ratio for China (3.4% of the USA's partnership strength) and the dominance of partnerships between high-income nations and a few high-burden nations suggest a core-periphery structure. This likely marginalizes endemic regions with limited research infrastructure, potentially creating blind spots in understanding local transmission dynamics, implementation challenges, and equitable intervention strategies. The strong transatlantic bridge (e.g., USA–France collaboration) underscores how shared resources and expertise amplify output, but this model must be expanded to foster more equitable and inclusive consortia.

Furthermore, the scarcity of African countries among the top publishing nations—despite their high disease burden—highlights a critical equity gap. This absence may stem from structural barriers, such as limited research funding and language biases in international indexing, and the lower visibility of regional journals. The underrepresentation of African institutions in collaboration networks risks perpetuating a knowledge imbalance, wherein local expertise, ecological insights, and implementation contexts are undervalued. This may lead to research agendas that are misaligned with regional priorities, inefficient resource allocation, and interventions that are less effective in local settings. Strengthening equitable partnerships with endemic regions in Africa and elsewhere is not only an ethical imperative but also essential for developing contextually appropriate and sustainable public health strategies.

The English-language restriction of our data may impact interpretations of cluster-specific collaboration dynamics. For example, while the USA cluster shows diverse global partnerships in English-language publications, Brazil may have stronger regional collaborations documented in Portuguese or Spanish (e.g., with Latin American countries) that are not captured here. Several Chinese institutions published their work in Mandarin-language journals, which may not be fully captured by our English-focused analysis, which could lead to an underestimation of the collaborative efforts of these institutions. To quantify this potential gap, we performed an additional search in two major Chinese bibliographic databases (Chinese Biomedical Literature Database (SinoMed) and WANFANG Database) using the Chinese term for chikungunya ("基孔肯雅热"). This search identified 350 unique Chinese-language publications on CHIKV (S9 Table). Among the 350 publications analyzed, all Chinese-language studies were published without coauthors from the United States or other countries. These findings align with trends in the Web of Science database, indicating that international collaboration in chikungunya research remains relatively limited in China. In addition to the potential underrepresentation of Chinese-language publications in our analysis, this limited collaboration is also due to the historically low incidence of the disease in China. This has led to a relatively weak domestic research base and thus limited motivation or infrastructure for extensive international cooperation. However, as chikungunya continues to spread globally, China also faces growing public health risks from this disease. In light of this, research needs in China are expected to increase significantly. Strengthening collaboration with the United States and other countries in this field could greatly enhance global efforts to prevent and control chikungunya. Transforming this imbalanced architecture into a robust and equitable global network will require systematic efforts to integrate non-English research into the global knowledge commons. Key steps include supporting the translation and indexing of key findings from regional journals and incorporating databases such as SciELO and CNKI into systematic reviews and bibliometric analyses.

### 4.4. Climate change and urbanization: Catalysts for CHIKV reemergence

This linkage is further corroborated by the tripling of publications in key endemic zones, such as India and Brazil, during the period of rapid research growth (2014–2020), which coincided with intensified climate impacts (Section 3.1; Section 3.2.1). The strong linkage between the keywords "climate change" and "chikungunya" (cooccurrence weight=16; Section 3.6) mirrors the tripling of publications in endemic zones such as India and Brazil during exponential growth phases (2014–2020), confirming that environmental drivers are research priorities. Critically, this paradigm shift is further evidenced by the emergence of "environmental health" as a central keyword in chikungunya research after 2022 (Section 3.6.4), reflecting intensified scientific focus on climate-driven vector expansion and its systemic health implications, implications spanning health system capacity, long-term patient care, and social equity.

The scope and impact of climate-sensitive diseases such as CHIKV are projected to intensify. Rising temperatures shorten mosquito incubation periods, expand vector habitats, and prolong transmission seasons [16]. Extreme weather events, such as flooding, can generate new breeding sites and disrupt public health infrastructure [17]. Dai et al. (2025) conducted a global assessment of current and future CHIKV transmission risk using optimized MaxEnt modeling. Their projections under various climate scenarios strongly support the expansion of climatically suitable areas for CHIKV transmission, thereby providing a mechanistic foundation for the observed research trends and the virus's potential for resurgence

[18]. Urban expansion, particularly unplanned settlements in tropical regions, fosters ideal conditions for Aedes proliferation, such as standing water, high human density, and poor sanitation [19]. Migration—both forced (e.g., due to conflict or disaster) and voluntary—further facilitates the spread of the virus across borders [20]. The inextricable linkage between climate change, unplanned urbanization, and CHIKV transmission underscores the urgent need for integrated planetary health strategies. As rising temperatures expand vector habitats and extreme weather events disrupt health infrastructure, proactive adaptation, such as predictive modeling of outbreak risk under future climate scenarios and climate-resilient urban planning, must become central to pandemic preparedness [18,21]. Critically, this paradigm shift is further evidenced by the emergence of "environmental health" as a central keyword in chikungunya research after 2022 (Section 3.6.4), reflecting an intensified scientific focus on climate-driven vector expansion and its systemic health implications. Mitigating CHIKV resurgence will require transboundary collaboration to address environmental drivers, strengthen surveillance in climate-vulnerable regions, and prioritize sustainable urbanization policies that disrupt Aedes breeding niches.

## 4.5. Limitations

This bibliometric analysis, while comprehensive, has several limitations. (1) Database and Language Bias: Our analysis relied exclusively on the Web of Science Core Collection (WoSCC). While WoSCC is a standard source for bibliometrics, it may not fully capture the scientific literature from all regions, particularly studies published in Scopus, PubMed, or regional databases (e.g., SciELO for Latin America, CNKI for China). This, combined with our inclusion of only English-language records, likely leads to an underrepresentation of research from non-English speaking endemic regions, such as parts of Latin America (Portuguese/Spanish), Africa (French/local languages), and Asia. This bias may affect the completeness of publication trends, collaboration network maps, and thematic analysis. To quantify one aspect of this potential gap, we conducted a supplementary search in two major Chinese bibliographic databases (SinoMed and WAN-FANG) using the Chinese term for chikungunya ("基孔肯雅热"). This search identified 350 unique Chinese-language publications (S9 Table), none of which involved international co-authorship, aligning with the limited collaboration observed for China in our WoS-based analysis. Future studies would benefit from multi-database searches and inclusive language strategies.". (2) Temporal Lag: Provisional 2025 data (through July 2025) may underrepresent late-year publications, affecting trend accuracy.

## 5. Conclusion

This 20-year bibliometric analysis delineates the evolution of research on chikungunya as the illness has developed from a niche tropical disease focus to a paradigm of a climate-sensitive pandemic threat. Three key findings emerged. First, the research output increased 7-fold after 2013, mirroring the global spread of CHIKV and its crystallization around the themes of climate-driven transmission, vaccine development, and arbovirus cocirculation. Second, collaboration networks remain anchored by high-income nations (the USA and France) and high-burden regions (Brazil and India), yet equitable partnerships—particularly with underresourced endemic zones—are underdeveloped. To foster equitable global CHIKV research partnerships, future initiatives should address not only resource gaps but also language barriers, for example, by supporting the translation of key research findings into non-English languages. This would ensure a more inclusive representation of regional research contributions and strengthen collaborative pandemic preparedness. Third, critical gaps persist: Despite advances in vaccine candidates (e.g., VLA1553) and vector control (e.g., Wolbachia), implementation barriers, diagnostic overlaps, and climate-adaptive surveillance hinder preparedness.

Moving forward, we advocate for research policies that 1) promote equitable partnerships to ensure that endemic region priorities are represented, 2) integrate climate-predictive modeling into surveillance and preparedness frameworks, and 3) bridge the translational gap between promising interventions (e.g., vaccines and Wolbachia) and real-world implementation. As CHIKV continues its expansion into temperate zones, this analysis provides a roadmap to align science, policy, and planetary health and to transform a reactive outbreak response into sustained pandemic resilience.

## Supporting information

**S1 Appendix. The complete dataset of 8,125 publications retrieved from the Web of Science Core Collection used in this bibliometric analysis.**
(ZIP)

**S1 Table. Consolidated and processed keyword list used for keyword co-occurrence and thematic analysis.**
(XLSX)

**S2 Table. Underlying data for Fig 2.**
(XLSX)

**S3 Table. Underlying data for Fig 3.**
(XLSX)

**S4 Table. Underlying data for Fig 4.**
(XLSX)

**S5 Table. Underlying data for Fig 5.**
(XLSX)

**S6 Table. Underlying data for Fig 6.**
(XLSX)

**S7 Table. Underlying data for Fig 7.**
(XLSX)

**S8 Table. Underlying data for Fig 8.**
(XLSX)

**S9 Table. Dataset of 350 publications related to chikungunya research from Chinese sources.**
(XLSX)

## Author contributions

**Conceptualization:** Xuetao Peng.

**Data curation:** Xiaoyi Xu.

**Formal analysis:** Yawei Liu.

**Funding acquisition:** Minyi He.

**Methodology:** Liming Ye.

**Validation:** Yawei Liu.

**Visualization:** Liming Ye.

**Writing – original draft:** Xiaoyi Xu, Xuetao Peng.

**Writing – review & editing:** Minyi He.

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
