## [Decision Letter · Decision Letter 0]

4 Sep 2025

Resurgent Threat in a Changing Climate: A 20-Year Bibliometric Analysis of Global Chikungunya Research Evolution and Pandemic Preparedness

Dear Dr. He,

Thank you for submitting your manuscript to PLOS Neglected Tropical Diseases. After careful consideration, we feel that it has merit but does not fully meet PLOS Neglected Tropical Diseases's publication criteria as it currently stands. Therefore, we invite you to submit a revised version of the manuscript that addresses the points raised during the review process.

Please submit your revised manuscript within 60 days Nov 03 2025 11:59PM. If you will need more time than this to complete your revisions, please reply to this message or contact the journal office at plosntds@plos.org. Please include the following items when submitting your revised manuscript:

We look forward to receiving your revised manuscript.

Kind regards,

Puneet Bhatt, MD, PhD

Academic Editor

Elvina Viennet

Section Editor

Shaden Kamhawi

co-Editor-in-Chief

Paul Brindley

co-Editor-in-Chief

**Additional Editor Comments:**

Reviewer #1:

Reviewer #2:

Reviewer #3:

**Journal Requirements:**

At this stage, the following Authors/Authors require contributions: Xiaoyi Xu, Liming Ye, Yawei Liu, Xuetao Peng, and Minyi He. Please ensure that the full contributions of each author are acknowledged in the "Add/Edit/Remove Authors" section of our submission form.

4) We note that your Data Availability Statement is currently as follows: "no". Please confirm at this time whether or not your submission contains all raw data required to replicate the results of your study. Authors must share the “minimal data set” for their submission. PLOS defines the minimal data set to consist of the data required to replicate all study findings reported in the article, as well as related metadata and methods (https://journals.plos.org/plosone/s/data-availability#loc-minimal-data-set-definition).

1) Please clarify all sources of financial support for your study. List the grants, grant numbers, and organizations that funded your study, including funding received from your institution. Please note that suppliers of material support, including research materials, should be recognized in the Acknowledgements section rather than in the Financial Disclosure

2) State the initials, alongside each funding source, of each author to receive each grant. For example: "This work was supported by the National Institutes of Health (####### to AM; ###### to CJ) and the National Science Foundation (###### to AM)."

3) State what role the funders took in the study. If the funders had no role in your study, please state: "The funders had no role in study design, data collection and analysis, decision to publish, or preparation of the manuscript."

4) If any authors received a salary from any of your funders, please state which authors and which funders..

6) Your current Financial Disclosure states, "The author(s) received no specific funding for this work.".

However, your funding information on the submission form indicates 3 funders .

Please indicate by return email the full and correct funding information for your study and confirm the order in which funding contributions should appear. Please be sure to indicate whether the funders played any role in the study design, data collection and analysis, decision to publish, or preparation of the manuscript.

7) Please send a completed 'Competing Interests' statement, including any COIs declared by your co-authors. If you have no competing interests to declare, please state "The authors have declared that no competing interests exist". Otherwise please declare all competing interests beginning with the statement "I have read the journal's policy and the authors of this manuscript have the following competing interests:"

8) We have noticed that you have uploaded Supporting Information files, but you have not included a list of legends. Please add a full list of legends for your Supporting Information files after the references list.

**Reviewers' Comments:**

Reviewer's Responses to Questions

**Key Review Criteria Required for Acceptance?**

**Methods**

-Are the objectives of the study clearly articulated with a clear testable hypothesis stated?

-Is the study design appropriate to address the stated objectives?

-Is the population clearly described and appropriate for the hypothesis being tested?

-Is the sample size sufficient to ensure adequate power to address the hypothesis being tested?

-Were correct statistical analysis used to support conclusions?

-Are there concerns about ethical or regulatory requirements being met?

Reviewer #1: 1. Introduction. There are several mentions of climate change, but no references are provided or really any effort to link climate change to the review, which is focused on the publication trends in CHIKV research. Revise the manuscript to include more introduction to the climate change link to CHIKV and describe how this is relevant to the literature review. This may be solved by moving the text from the discussion on page 28 beginning “This aligns with the epidemiological trajectory…” and ending with “… perfect ecological storm for the virus’s re-establishment.” Alternatively, I suggest removing the climate change discussion. It is not strictly necessary to the central messages of the manuscript.

2. Section 2.4. For transparency and reproducibility, I suggest that the merged terms, etc be described in a supplementary table.

3. There is a bias in the data, in that it only discusses English language articles but displays them globally. This is the biggest issue with this manuscript. This is acknowledged in the limitations section, but should be acknowledged in the methods.

Reviewer #2: data analysis and visualisation has not been discussed.

Reviewer #3: -Are the objectives of the study clearly articulated with a clear testable hypothesis stated? YES

-Is the study design appropriate to address the stated objectives? YES

-Is the sample size sufficient to ensure adequate power to address the hypothesis being tested? YES

-Were correct statistical analysis used to support conclusions? YES

**Results**

-Does the analysis presented match the analysis plan?

-Are the results clearly and completely presented?

-Are the figures (Tables, Images) of sufficient quality for clarity?

Reviewer #1: 4. It would be useful to reiterate the English language bias in section 3.2.1, as nodes of articles from Brazil, India, and China may have more non-English publications, where national contributions are expressed. Note that English is the language of the UK and US. The authors draw conclusions on international relationships, without more clearly acknowledging the bias.

5. The statements in the cluster specific dynamics are affected by the English only bias. While the USA cluster has diverse global partnerships, it is unclear if Brazil has better international partnerships in Portuguese or Spanish language. This also applies to the Other Countries, including the Asia-Pacific. It is not surprising to me that Chinese institutions are publishing less in English language journals than US institutions. This needs to be included in the discussion.

Reviewer #2: Major

The results and discussions are disjointed. Since this is a bibiometric analysis, I anticipate the discussion will highlight the findings, significance and limitations from this analysis, rather than discussing the epidemiology (section 4.1), vaccine development (4.2) etc.

There are many sections of over-interpretation of data. One example is in 4.3 Public Health Strategies and Their Shortcomings, the authors discussed the decline in “"Emergency Response and Crisis Management" keywords (16.2% to 7.2% from 2016-2017 to 2024-2025) within public health research suggests a problematic transition away from reactive capacity, despite the ongoing threat”

There are many factors that can contribute to the decline in certain keywords, and as there is no objective evaluation (qualitative/quantitative measure) presented here.

The selection of English language publication alone in this analysis is a key limitation when assessing equitable partnerships.

Minor

Page 12, under 3.2.1 National Contributions: “The United States (USA) dominated with 2,622 publications, constituting(2.9% of the total…

Is this 29% rather than 2.9%?

Reviewer #3: -Does the analysis presented match the analysis plan? YES

-Are the results clearly and completely presented? YES

-Are the figures of sufficient quality for clarity? YES

**Conclusions**

-Are the conclusions supported by the data presented?

-Are the limitations of analysis clearly described?

-Do the authors discuss how these data can be helpful to advance our understanding of the topic under study?

-Is public health relevance addressed?

Reviewer #1: 6. At the bottom of page 33, there is a typo. There needs to be a space after the period : “implications.Mitigating”

7. In the conclusions, when discussing equitable partnerships, perhaps language should also be discussed.

Reviewer #2: There is overcalling in the conclusion “As CHIKV continues its expansion into temperate zones, this analysis provides a roadmap to align science, policy, and planetary health—transforming reactive outbreak response into sustained pandemic resilience.”

This analysis has numerous limitations and factors that could influence the findings, and the result is only a publication snapshot which has been interpreted.

Reviewer #3: -Are the conclusions supported by the data presented? YES

-Are the limitations of analysis clearly described? YES

-Do the authors discuss how these data can be helpful to advance our understanding of the topic under study? YES

-Is public health relevance addressed? YES

**Editorial and Data Presentation Modifications?**

Reviewer #1: Minor Revision

Reviewer #2: (No Response)

Reviewer #3: “Minor Revision”

**Summary and General Comments**

Reviewer #1: This is a good review. The greatest weakness is the English language bias and that needs to be addressed frequently in all sections. If that is done, then it will be useful to the larger scientific community.

Reviewer #2: (No Response)

Reviewer #3: The manuscript by Xiaoyi et al., titled Resurgent Threat in a Changing Climate: A 20-Year Bibliometric Analysis of Global Chikungunya Research Evolution and Pandemic Preparedness is a comprehensive bibliometric analysis of global chikungunya virus (CHIKV) research over the past two decades, with a focus on thematic evolution, collaboration networks, and alignment with pandemic preparedness. The study is relevant, especially in light of climate-driven arboviral resurgence. The methodology is robust, and the findings offer valuable insights into research priorities, gaps, and future directions. The text is well-organized and data-rich, though several areas could benefit from clarification and refinement. Below are some comments to improve clarity and contextual depth, to make it suitable for publication.

PLOS authors have the option to publish the peer review history of their article (what does this mean? ). If published, this will include your full peer review and any attached files.

**Do you want your identity to be public for this peer review?** For information about this choice, including consent withdrawal, please see our Privacy Policy .

Reviewer #1: No

Reviewer #2: **Yes:** Chuan Kok Lim

Reviewer #3: **Yes:** DEMANOU Maurice

**Figure resubmission:**
---

## [Decision Letter · Decision Letter 1]

10 Oct 2025

Thank you for submitting your manuscript to PLOS Neglected Tropical Diseases. After careful consideration, we feel that it has merit but does not fully meet PLOS Neglected Tropical Diseases's publication criteria as it currently stands. Therefore, we invite you to submit a revised version of the manuscript that addresses the points raised during the review process.

Please submit your revised manuscript within 30 days Nov 09 2025 11:59PM. If you will need more time than this to complete your revisions, please reply to this message or contact the journal office at plosntds@plos.org. Please include the following items when submitting your revised manuscript:

* A rebuttal letter that responds to each point raised by the editor and reviewer(s). You should upload this letter as a separate file labeled 'Response to Reviewers '. This file does not need to include responses to any formatting updates and technical items listed in the 'Journal Requirements' section below.

* A marked-up copy of your manuscript that highlights changes made to the original version. You should upload this as a separate file labeled 'Revised Manuscript with Track Changes '.

* An unmarked version of your revised paper without tracked changes. You should upload this as a separate file labeled 'Manuscript '.

We look forward to receiving your revised manuscript.

Kind regards,

Puneet Bhatt, MD, PhD

Academic Editor

co-Editor-in-Chief

Paul Brindley

co-Editor-in-Chief

**Reviewers' comments:**

**Key Review Criteria Required for Acceptance?**

**Methods:**

-Are the objectives of the study clearly articulated with a clear testable hypothesis stated?

-Is the study design appropriate to address the stated objectives?

-Is the population clearly described and appropriate for the hypothesis being tested?

-Is the sample size sufficient to ensure adequate power to address the hypothesis being tested?

-Were correct statistical analysis used to support conclusions?

-Are there concerns about ethical or regulatory requirements being met?

Reviewer #1: The methods are clearly presented and appropriate with the revised objectives.

Reviewer #3: The authors have provided a more transparent account of their inclusion/exclusion criteria, particularly in relation to duplicate handling, non-English records, and keyword normalization. The clarification of the semi-automated process using the bibliometrix package, followed by manual verification, is appropriate and enhances reproducibility. The integration of non-English publications and the rationale for excluding generic terms such as “review” and “study” are also well justified and improve the global representativeness of the dataset.

Reviewer #4: (No Response)

**Results:**

-Does the analysis presented match the analysis plan?

-Are the results clearly and completely presented?

-Are the figures (Tables, Images) of sufficient quality for clarity?

Reviewer #1: The results are more in line with the revised objectives of global preparedness. Figures are clear.

Reviewer #3: The analysis presented aligns well with the stated methodological framework. The authors have clarified key procedures (including duplicate handling, keyword normalization, and inclusion of non-English records) which now reflect a coherent and transparent approach consistent with the original plan.

The results are clearly and comprehensively presented. The authors have strengthened the interpretation of thematic trends, collaboration networks, and vaccine-related developments.

The figures and tables are of sufficient quality for clarity. The relocation of legends in Figures 6–8 enhances readability.

Reviewer #4: (No Response)

**Conclusions:**

-Are the conclusions supported by the data presented?

-Are the limitations of analysis clearly described?

-Do the authors discuss how these data can be helpful to advance our understanding of the topic under study?

-Is public health relevance addressed?

Reviewer #1: This revision is a significant improvement. The English language concerns are better addressed. However, the conclusions of translating English language into other languages. However, this conclusion seems like a one-way exchange. The authors point to the low volume of American-Chinese collaborations, but give no insight into the volume of Chikungunya publications in the Chinese literature. I suggest on Page 34, after the authors state: "Similarly, Chinese institutions—though active in CHIKV research—publish a portion of their work in Mandarin-language journals, leading to potential underestimation of their collaboration frequency in our English-focused analysis." the authors should add some additional information, such as the number of articles that found in Chinese language biomedical bibliographic databases when seached for "Chikungunya". From databases such as the Chinese Biomedical Literature Database or WANFANG database. Also, I would encourage the authors to be more ambitious in their conclusions on strengthening US-Chinese research. For example, calling for joint publication of articles in both Chinese and English language.

Reviewer #3: The conclusions are well supported by the data presented. The authors have drawn appropriate inferences from their bibliometric analysis, and the revised discussion sections reflect a balanced interpretation of trends, collaboration patterns, and thematic evolution.

The authors have clearly described the limitations of their analysis, including language bias, indexing constraints, and structural barriers affecting representation from endemic regions.

The public health implications are explicitly addressed, particularly in relation to outbreak preparedness, vaccine deployment, and the need for equitable research partnerships.

Reviewer #4: (No Response)

**Editorial and Data Presentation Modifications?**

Reviewer #1: Minor Revisions.

There is a typo on Page 42 where it says "preparedness.Third" there should be a space after the period.

Reviewer #3: Accept

Reviewer #4: (No Response)

**Summary and General Comments:**

Reviewer #1: Much improved article, but I believe that English language Chikungunya research community would benefit from a more ambitious conclusion section, where the authors propose how to strengthen the global network, particularly how English speaking regions can engage with non-English speaking regions, with particular emphasis on China, since the authors are from Chinese institutions.

Reviewer #3: Overall, I find the authors’ revisions satisfactory and believe they have addressed the concerns raised. I have no further major comments at this stage.

Reviewer #4: The authors have made substantial improvements in this revised version. The manuscript is now more methodologically transparent, globally representative, and tightly focused on interpreting bibliometric findings rather than reiterating general epidemiological knowledge. Most of the previous concerns have been adequately addressed. I recommend acceptance after minor revisions.

1. Page 35, under 4.4 Climate Change and Urbanization: Catalysts for CHIKV Re-emergence

What does "systemic health" mean? Dose it refer to the overall well-being and functional integrity of the entire body, ecology, or other aspects? Please provide an explanation.

2. The manuscript needs to be edited by a native English speaker before it can be considered acceptable for publication.

PLOS authors have the option to publish the peer review history of their article (what does this mean? ). If published, this will include your full peer review and any attached files.

**Do you want your identity to be public for this peer review?** For information about this choice, including consent withdrawal, please see our Privacy Policy .

Reviewer #1: No

Reviewer #3: **Yes:** DEMANOU Maurice

Reviewer #4: No

**Figure resubmission:**

**Reproducibility:**To enhance the reproducibility of your results, we recommend that authors of applicable studies deposit laboratory protocols in protocols.io, where a protocol can be assigned its own identifier (DOI) such that it can be cited independently in the future. Additionally, PLOS ONE offers an option to publish peer-reviewed clinical study protocols. Read more information on sharing protocols at https://plos.org/protocols?utm_medium=editorial-email&utm_source=authorletters&utm_campaign=protocols

---

## [Decision Letter · Decision Letter 2]

23 Dec 2025

Response to Reviewers
Revised Manuscript with Track Changes
Manuscript

Shaden Kamhawi

co-Editor-in-Chief

Paul Brindley

co-Editor-in-Chief

**Journal Requirements:**

Please provide an Author Summary. This should appear in your manuscript between the Abstract (if applicable) and the Introduction, and should be 150-200 words long. The aim should be to make your findings accessible to a wide audience that includes both scientists and non-scientists. Sample summaries can be found on our website under Submission Guidelines:

**Reviewers' comments:**

**Key Review Criteria Required for Acceptance?**

**Methods**

-Are the objectives of the study clearly articulated with a clear testable hypothesis stated?

-Is the study design appropriate to address the stated objectives?

-Is the population clearly described and appropriate for the hypothesis being tested?

-Is the sample size sufficient to ensure adequate power to address the hypothesis being tested?

-Were correct statistical analysis used to support conclusions?

-Are there concerns about ethical or regulatory requirements being met?

Reviewer #1: I would like to commend the authors on the second revision of this manuscript. Objectives are clearly stated, design is appropriate, sample size is descibed and appropriate. I appreciate the effort to get to this excellent manuscript.

Reviewer #4: 1. Justify the criterion of " Justify the criterion of "minimum 50 occurrences" (e.g., whether sensitivity verification was performed)." (e.g., whether sensitivity verification was performed).

2. In Analytical Framework,A bibliometric analysis was conducted employing R 4.3.3, however, In Data Analysis and Visualization, Bibliometric data were analyzed using …… R (version 4.4.3). You should ensure the version is consistent throughout your paper and reflects the most appropriate one used for the analysis.

3. Relying solely on Web of Science may miss literatures from Scopus/PubMed (especially non-English regions).

**Results**

-Does the analysis presented match the analysis plan?

-Are the results clearly and completely presented?

-Are the figures (Tables, Images) of sufficient quality for clarity?

Reviewer #1: Analysis plan is clear, results are clear in figures and text.

Reviewer #4: (No Response)

**Conclusions**

-Are the conclusions supported by the data presented?

-Are the limitations of analysis clearly described?

-Do the authors discuss how these data can be helpful to advance our understanding of the topic under study?

-Is public health relevance addressed?

Reviewer #1: Conclusion are supported by the data. Limitations are analyzed and well addressed. There is a good discussion on how to advance, with particular emphasis on public health.

Reviewer #4: (No Response)

**Editorial and Data Presentation Modifications?**

Reviewer #1: I did notice three unusual word usages. On page 8, "coword" should be "co-word" and "biclustering" should be "bi clustering" or "bi-clustering" (I defer to PLOS editoral staff on the correct word usage). On page 21 "Cocitation" should be "Co-citation".

Reviewer #4: (No Response)

**Summary and General Comments**

Reviewer #1: I want to commend the authors on putting in the additional effort to improve the manuscript so that it is a strong contribution to Chikungunya literature.

Reviewer #4: (No Response)

PLOS authors have the option to publish the peer review history of their article (what does this mean? ). If published, this will include your full peer review and any attached files.

**Do you want your identity to be public for this peer review?** For information about this choice, including consent withdrawal, please see our Privacy Policy .

Reviewer #1: **Yes:** H. Carl Gelhaus

Reviewer #4: **Yes:** Guojun Cai

**Figure resubmission:**

**Reproducibility:** To enhance the reproducibility of your results, we recommend that authors of applicable studies deposit laboratory protocols in protocols.io, where a protocol can be assigned its own identifier (DOI) such that it can be cited independently in the future. Additionally, PLOS ONE offers an option to publish peer-reviewed clinical study protocols. Read more information on sharing protocols at https://plos.org/protocols?utm_medium=editorial-email&utm_source=authorletters&utm_campaign=protocols

---

## [Editor Report · Decision Letter 3]

20 Jan 2026

Dear Dr. He,

We are pleased to inform you that your manuscript 'Resurgent Threat in a Changing Climate: A 20-Year Bibliometric Analysis of Global Chikungunya Research Evolution and Pandemic Preparedness' has been provisionally accepted for publication in PLOS Neglected Tropical Diseases.

Best regards,

Sujatha Sunil, PhD

Section Editor

Sujatha Sunil

Section Editor

Shaden Kamhawi

co-Editor-in-Chief

Paul Brindley

co-Editor-in-Chief

---

## [Editor Report · Acceptance letter]

Dear Dr. He,

We are delighted to inform you that your manuscript, "

Resurgent Threat in a Changing Climate: A 20-Year Bibliometric Analysis of Global Chikungunya Research Evolution and Pandemic Preparedness," has been formally accepted for publication in PLOS Neglected Tropical Diseases.

Best regards,

Shaden Kamhawi

co-Editor-in-Chief

Paul Brindley

co-Editor-in-Chief
